# Hearing Impairment Overview in Africa: the Case of Cameroon

**DOI:** 10.3390/genes11020233

**Published:** 2020-02-22

**Authors:** Edmond Wonkam Tingang, Jean Jacques Noubiap, Jean Valentin F. Fokouo, Oluwafemi Gabriel Oluwole, Séraphin Nguefack, Emile R. Chimusa, Ambroise Wonkam

**Affiliations:** 1Division of Human Genetics, Department of Pathology, University of Cape Town, Cape Town 7925, South Africa; wonkamedmond@yahoo.fr (E.W.T.); oluwafemi.oluwole@uct.ac.za (O.G.O.); emile.chimusa@uct.ac.za (E.R.C.); 2Centre for Heart Rhythm Disorders, South Australian Health and Medical Research Institute (SAHMRI), University of Adelaide and Royal Adelaide Hospital, Adelaide 5000, Australia; noubiapjj@yahoo.fr; 3ENT unit, Bertoua Regional Hospital, P.O. Box 40, Bertoua, Cameroon; valentin.fokouo@gmail.com; 4Department of Paediatrics, Faculty of Medicine and Biomedical Sciences, University of Yaoundé I, Yaoundé 1364, Cameroon; seraphin_nguefack@yahoo.fr; 5Paediatrics unit, Gynaeco-Obstetric and Paediatric Hospital, Yaoundé 4362, Cameroon; 6Department of Medicine, University of Cape Town, Cape Town 7925, South Africa

**Keywords:** hearing impairment, prevalence, etiologies, genetics, Cameroon, Africa

## Abstract

The incidence of hearing impairment (HI) is higher in low- and middle-income countries when compared to high-income countries. There is therefore a necessity to estimate the burden of this condition in developing world. The aim of our study was to use a systematic approach to provide summarized data on the prevalence, etiologies, clinical patterns and genetics of HI in Cameroon. We searched PubMed, Scopus, African Journals Online, AFROLIB and African Index Medicus to identify relevant studies on HI in Cameroon, published from inception to 31 October, 2019, with no language restrictions. Reference lists of included studies were also scrutinized, and data were summarized narratively. This study is registered with PROSPERO, number CRD42019142788. We screened 333 records, of which 17 studies were finally included in the review. The prevalence of HI in Cameroon ranges from 0.9% to 3.6% in population-based studies and increases with age. Environmental factors contribute to 52.6% to 62.2% of HI cases, with meningitis, impacted wax and age-related disorder being the most common ones. Hereditary HI comprises 0.8% to 14.8% of all cases. In 32.6% to 37% of HI cases, the origin remains unknown. Non-syndromic hearing impairment (NSHI) is the most frequent clinical entity and accounts for 86.1% to 92.5% of cases of HI of genetic origin. Waardenburg and Usher syndromes account for 50% to 57.14% and 8.9% to 42.9% of genetic syndromic cases, respectively. No pathogenic mutation was described in *GJB6* gene, and the prevalence of pathogenic mutations in *GJB2* gene ranged from 0% to 0.5%. The prevalence of pathogenic mutations in other known NSHI genes was <10% in Cameroonian probands. Environmental factors are the leading etiology of HI in Cameroon, and mutations in most important HI genes are infrequent in Cameroon. Whole genome sequencing therefore appears as the most effective way to identify variants associated with HI in Cameroon and sub-Saharan Africa in general.

## 1. Introduction

Hearing impairment (HI) is considered disabling when the loss of hearing is greater than 40 dB in the better-hearing ear in adults (15 years or older) or greater than 30 dB in the better-hearing ear in children (0 to 14 years) [1]. The World Health Organization (WHO) estimated in 2018 that approximately 466 million people globally live with disabling HI (6.1% of the world population), of whom 34 million are children and 49 million live in sub-Saharan Africa [1]. HI affects up to 6 per 1000 live births in sub-Saharan Africa, with a lower incidence of about 1 per 1000 live births in developed countries [2,3].

According to WHO, HI can be classified as mild, moderate, severe or profound when the pure tone average ranges from 26 to 40 dB, 41 to 60 dB, 61 to 80 dB or is over 81 dB, respectively [1]. HI can be due to environmental or genetic causes, and in many cases it is not possible to establish a definite etiology [4,5]. Environmental factors such as meningitis, measles or ototoxicity are the leading causes of HI in low- and middle-income countries, while their burden is lower in high income countries [5,6,7,8]. This is attributed to poor healthcare systems that are not always adequately equipped to prevent, screen and manage causes of HI [7]. Genetic factors contribute to 30% to 50% of HI cases in sub-Saharan Africa [7]. HI of genetic origin can be syndromic or non-syndromic, depending on whether it is associated with additional abnormalities in other organs or not [7,9,10]. Non-syndromic hearing impairment (NSHI) accounts for 70% of hereditary hearing loss [11]. Over 400 syndromes with HI have been described, including Waardenburg syndrome, branchiootorenal syndrome, Usher syndrome, Pendred syndrome, keratitis–ichthyosis–deafness syndrome and Alport syndrome [7,11,12].

*GJB2* (on chromosome 13q12) is the most commonly associated gene with NSHI in European and Asian populations and accounts for almost 50% of cases [11,13,14]. The most common *GJB2* mutation is c.35delG, which represents about 70% of *GJB2* mutated alleles in those populations [7,15]. Other mutations are prevalent in specific populations, including 167delC among the Ashkenazim in Israel and p.R143W in Ghana [16,17,18].

Regardless of its etiology, uncorrected HI has sequelae [19]. Undetected and untreated hearing loss can result in poor reading performance, poor communication skills and poor speech production [19,20]. Educational intervention is insufficient to completely remediate these deficiencies. In contrast, early auditory intervention is effective, whether through amplification, otologic surgery or cochlear implantation [19].

Some systematic reviews have estimated the global and the regional burden of HI [21,22]; however, there is a lack of data on national estimates of the prevalence of this condition in many countries, especially in Africa. Therefore, there is a call to action for each country to estimate its national burden and to develop specific programs for the prevention and the management of HI. The aim of the present review was to use a systematic approach to provide summarized data on the prevalence, etiologies, clinical patterns and genetics of HI in Cameroon, a sub-Saharan African country.

## 2. Materials and Methods

This review is reported in accordance with the Preferred Reporting Items for Systematic Review and Meta-Analysis (PRISMA) statement [23] and is registered in the International Prospective Register of Systematic Reviews (PROSPERO, registration number: CRD42019142788).

### 2.1. Selection Criteria 

We included observational studies published from inception to 31 October, 2019 that report data on the prevalence, etiologies, clinical characteristics or genetics of HI in Cameroon. For duplicate studies, the most comprehensive and/or recent article with the largest sample size was considered. Qualitative studies, letters to the editor, reviews and commentaries were excluded. Studies with either unavailable full text or missing key data that could not be accessed after a reasonable request from corresponding authors, or before the end of the data extraction process, were also excluded.

### 2.2. Search Strategy

We searched PubMed, Scopus, and African-specific databases (African Journals Online, AFROLIB and African Index Medicus) for relevant articles. Keywords used for the search included: “hearing loss”, “hearing impairment”, “deafness”, “deaf” and “Cameroon”. All the search strategies are represented in the Appendix A.

The reference lists of all eligible studies and of relevant literature reviews were screened, and specific researchers active in the field of hearing loss in Cameroon were contacted to identify additional sources of information.

### 2.3. Selection of Studies

Titles and abstracts obtained from searches were imported into the software Zotero, version 5.0.64, for the removal of any duplicates. Regarding our inclusion and exclusion criteria, one author (EWT) screened unduplicated titles and abstracts before reviewing the full text of all selected studies for final inclusion. A second author (JJN) verified that the study screening and selection process was performed correctly. Any disagreement between the two authors was solved through discussion and consensus.

### 2.4. Data Extraction Process

Using a predesigned data extraction sheet, one researcher (EWT) extracted data from relevant studies. A second researcher (JJN) checked the accuracy of the data extraction process, with any discrepancy resolved through discussion and consensus. Extracted data included: the last name of first author, the year of publication, the region(s) where the study was conducted, setting (hospital, school for the deaf, community), study design, data collection (prospective versus retrospective), study population, proportion of males, mean or median age, age range, sample size, tool used to diagnose HI, number of cases of HI (for prevalence studies), proportions of different types of hearing loss (sensorineural, conductive, mixed), proportions of different levels of hearing loss (mild, moderate, severe, profound), distribution of etiologies, inheritance patterns for genetic cases, clinical patterns with details (syndromic versus non-syndromic), data on molecular testing including genotyping method, targeted genes or mutations and pathogenic variants found. For some studies, relevant proportions were calculated from their raw data. Data were summarized narratively.

### 2.5. Assessment of Methodological Quality

Two investigators (EWT and ERC) assessed the risk of bias and the quality of included studies using the quality of genetic studies (Q-Genie) tool developed by Sohani et al. [24] for genetic studies and the risk of bias assessment tool for prevalence studies developed by Hoy et al. [25] for the other studies. Discrepancies were solved by discussion and consensus. 

## 3. Results

### 3.1. The Review Process

Initially, 333 records were identified through database searches. After the removal of duplicates, titles and abstracts of 289 studies were screened, of which 255 records were excluded. Full texts of the remaining 34 records and of papers identified through other sources were scrutinized for final inclusion. A total of 17 articles were judged to be eligible and were included in the review (Figure 1).

### 3.2. Characteristics of Included Studies 

The characteristics of the included studies are summarized in Table 1. The majority of the studies were school- and/or hospital-based. Patients included in these studies were globally recruited from 9 of the 10 administrative regions of Cameroon and were recruited from both urban and rural areas. The male proportion ranged from 0% to 74.4%, and the mean age ranged from 3.5 to 43.2 years. Pure tone audiometry (PTA) was the most frequent tool used to diagnose HI. The study quality was high, moderate and low in ten [26,27,28,29,30,31,32,33,34,35], three [36,37,38] and four [39,40,41,42] studies, respectively.

### 3.3. Prevalence of Hearing Impairment in Cameroon

None of the studies included in this review reported on the national prevalence of HI in Cameroon. However, the regional estimates reported by these studies give an insight into the public health burden of this condition in Cameroon. In 2013, a community-based study was carried out in a health district located in the Northwest region of Cameroon and included the general population of that area. A total of 3567 individuals were recruited and screened through PTA and otoacoustic emission test. They were aged 0 to over 80 years old, with a male proportion of 40.8%. The overall prevalence of HI in this population was estimated at 3.6% (95% CI: 2.8–4.6) [32]. The prevalence was relatively low at 1.1% among children (<18 years), was high as 6.5% among adults (≥18 years old), and rose rapidly to a level of 14.8% in participants aged 50 years and more [32]. The prevalence of HI in this health district seems to be greater than the overall prevalence in the Northwest region of Cameroon, which was estimated at 0.9% in a study including 18,878 participants from that region and where HI was diagnosed on a self-reported base [42].

In a retrospective cross-sectional study conducted at the Yaoundé teaching hospital, the authors included patients who visited the ear nose and throat (ENT) unit from January 2010 to July 2014 for sensorineural emergencies, which includes sudden sensorineural hearing loss, Bell’s palsy and acute vertigo [39]. A total of 22 patients were included in the study, with a mean age of 43.2 ± 17 (range 16–66) years. The prevalence of sudden sensorineural hearing loss, which was defined as an NSHI occurring within 72 h, was estimated at 9.1% (2 out of 22) in this group of patients [39].

### 3.4. Audiometric Characteristics of Hearing Impairment in Cameroon

In Cameroon, sensorineural HI is the most frequent pathophysiological type, and is accounted for in 61.7% to 94.4% of all HI cases, while mixed and conductive HI are found in 5.6% to 20% and 0% to 18.3% of cases, respectively [26,27,29]. HI tend to be more severe in school settings, where profound HI (≥81 dB) accounts for 93.5% to 98.2% of cases [27,29,36], than in community settings, where profound HI is observed in only 9% of cases, the majority (76%) being moderate HI (41–60 dB) [32]. Bilateral HI represent 36% to 100% of cases [26,27,29,36]; and in the case-control study performed in a referral hospital of Yaoundé, HI was left-sided and right-sided in 43% and 21% of participants, respectively [26].

### 3.5. Etiologies of Hearing Impairment in Cameroon

Environmental factors are the leading causes of HI in Cameroon, and account for 52.6% to 62.2% of HI cases [27,32]. Hereditary HI is responsible for 0.8% to 14.8% of cases, and in 32.6% to 37% of HI cases the origin remains unknown (Table 2) [27,32].

In a cohort including 582 hearing-impaired children from 7 of the 10 administrative regions of Cameroon, meningitis was the leading environmental etiology and was implicated in 34.4% of HI cases (Table 2). Infectious diseases that are preventable by vaccination (meningitis, measles, rubella and mumps) represented 41.3% of cases [27]. A cross-sectional study performed in 2014 emphasized the link between rubella infection and deafness [40]. The authors recruited 320 children from two different schools in the Northwest region of Cameroon, including one school for the deaf. They found that hearing-impaired children were seven times more likely to have positive rubella IgG serology (48.7%) than children with normal hearing (7.4%; *p* < 0.0001) [40]. 

In a community-based study undertaken in 2013 in Fundong health district, impacted wax and age-related HI were the most frequent environmental etiologies and accounted for 31.5% and 22.8% of HI cases, respectively (Table 2). Age-related HI was particularly prevalent in elderly patients and was implicated in 31% of HI cases in patients aged 50 years or more [32]. Severe malaria has been identified as a cause of deafness in a retrospective cross-sectional study performed in 2004 [41]. The authors assessed the outcome of severe malaria in 387 patients admitted and treated in a regional hospital of the East region of Cameroon. Among the 317 patients who recovered, neurological sequelae were observed in six patients, of which three had deafness [41].

Ototoxicity due to anti-tuberculosis drugs also plays an important role in the etiology of HI in Cameroon. A longitudinal study performed in two hospitals, located in two different regions of Cameroon, assessed the outcome of anti-tuberculosis treatment in multidrug-resistant tuberculosis (MDR-TB) patients. Patients were treated with a standardized 12-month drug regimen, including gatifloxacin, clofazimine, prothionamide, ethambutol and pyrazinamide throughout, supplemented by kanamycin and isoniazid during an intensive phase of a minimum of 4 months. Participants underwent audiometry testing at baseline and after 4 months treatment. A total of 150 participants were recruited, and their mean age was 33.7 years (range 17–68). Among the 106 patients with audiometry data available, 46 (43.4%) presented HI after a 4 month follow-up [37]. A similar study performed in nine African countries (Cameroon, Burkina Faso, Burundi, Benin, Democratic Republic of Congo, Central Africa Republic, Ivory Coast, Niger, and Rwanda), assessed the adverse effects of anti-tuberculosis drugs in MDR-TB patients, treated with a standardized 9-month regimen, including moxifloxacin, clofazimine, ethambutol and pyrazinamide throughout, supplemented by kanamycin, prothionamide and high-dose isoniazid during an intensive phase of a minimum of 4 to a maximum of 6 months. Of the 491 patients with audiometry results available at both month 0 and month 4, 56 (11.4%) had severe hearing deterioration at month 4 [38]. 

With reference to a case-control study undertaken in a referral hospital in Yaoundé, HIV infection seems to be one of the etiological factors of HI in the Cameroonian population. Included in the aforementioned study were 90 HIV-infected individuals divided in three subgroups: 30 highly active antiretroviral therapy (HAART)-naive patients, 30 patients receiving first-line HAART and 30 patients receiving second-line HAART, as well as 90 apparently healthy participants as controls. The prevalence of HI in the HIV-positive group was higher than the HIV negative group (27.2% vs. 5.6%; *p* = 0.04) [26]. Additionally, HIV-positive patients had a significant increase in pure tone averages when compared with the HIV-negative patients. No significant difference was found in terms of pure tone averages and susceptibility to hearing loss between HAART-naive patients and those receiving HAART [26].

### 3.6. Hearing Impairment of Genetic Origin

#### 3.6.1. Clinical Patterns

Concerning hereditary HI, NSHI is the most prevalent clinical entity in Cameroon, and accounts for 86.1% to 92.5% of cases, while syndromic HI represents 7.5% to 13.9% of cases [27,29].

Waardenburg syndrome (WS) is the most frequent syndromic HI in the Cameroonian population; it represents 1% of all HI cases, 4.3% to 7% of HI of genetic origin, and 50% to 57.14% of genetic syndromic cases [27,29,35]. All cases of WS described in the Cameroonian population are type 2 (without dystopia canthorum). Clinical signs included: congenital severe to profound sensorineural HI, hypopigmentation of the skin (usually on the trunk), premature canitis, isochromic sapphire-blue eyes and in some cases heterochromia iridis that can be segmental or complete (Figure 2) [27,29,35].

Usher syndrome is the second most frequent syndromic HI in the Cameroonian population. It accounts for 8.9% to 42.9% of all syndromic HI cases [27,29]. Three cases of type 2 Usher syndrome have been described with clinical signs of retinitis pigmentosa, including night vision impairment and constricted visual field present in affected patients, in addition to HI [29]. A case of type 1 Usher syndrome was reported, and clinical signs included those of type 2, associated with vestibular dysfunction [27].

Two cases of keratitis–ichthyosis–deafness (KID) syndrome were reported. PTA in these patients showed bilateral profound sensorineural HI. Physical examination revealed generalized ichtyosis and erythrokeratodermia, palmoplantar keratoderma, rippled hyperkeratotic plaques on the knees, elbows and ankles (reducing the mobility of the affected joints), hypotrichosis, alopecia, and hyperkeratosis lesions in the external auditory canal. Ophthalmologic examination revealed a mild vascularizing keratitis which explained photophobia and reduced visual acuity. Oral examination showed dental dysplasia, and histopathological examination of the skin revealed an acanthotic dyskeratosis (Figure 3) [27,30].

One case of oculo-auriculo-vertebral (OAV) spectrum, presenting with unilateral facial hypoplasia, anti-mongoloid slant of the palpebral fissures, microtia and preauricular tags, a slight degree of mental retardation, vertebral anomalies and deafness, and a case of Pendred syndrome that presented with a postlingual progressive sensorineural deafness and hypothyroid goitre have also been described [27].

#### 3.6.2. Inheritance Pattern

Based on pedigree analysis, studies reported autosomal recessive inheritance to be the most frequently observed pattern of inheritance, accounting for 82.8% to 87.2% of cases of HI of genetic origin. Autosomal dominant and mitochondrial inheritance were less frequent and were observed in 9.3% to 10.7% and 0% to 6.5% of cases, respectively. Pedigree-based consanguinity was estimated at 6.5% to 13.1% of cases of hereditary HI in the Cameroonian population (Appendix A) [27,29].

#### 3.6.3. Gene Variants and Hearing Impairment

##### Non-syndromic Hearing Impairment

Several studies assessed the contribution of known HI genes to HI in the Cameroonian population. Mutations in *GJB2* and *GJB6* genes have a prevalence close to zero in the Cameroonian population. In 2011, a cohort of 70 deaf children from the Far-North region of Cameroon were screened for mutations in *GJB2* and *MTRNR1* (mitochondrial DNA) genes, by direct sequencing. No mutations were found in *GJB2*, and a *MTRNR1* variant of unknown pathogenicity (m.G1462T) was present in one patient [36]. Additionally, a cohort of 180 Cameroonians with non-syndromic deafness of either putative genetic origin or unknown origin were screened for *GJB2* mutations in 2014. A pathogenic mutation, c.424_426delTTC (p.F142del), and a mutation of uncertain significance, c.499G>A (p.V167M), were both detected in one patient, in the heterozygous form [33]. Subsequently, 75 Cameroonian patients with non-syndromic deafness were analyzed through direct Sanger sequencing of the entirety of the coding regions of *GJB6* gene and *GJA1* pseudo-gene; the large-scale *GJB6*-D3S1830 deletion was also investigated. No pathogenic mutations were detected in either *GJB6* or *GJA1*, nor was the *GJB6*-D3S1830 deletion detected [31]. 

Recently, 29 non-syndromic hearing-impaired families, showing a clear segregation of the disease within the family, with at least two affected family members and with strong evidence of non-environmental etiology, were recruited. They were screened for mutations in the coding exon (exon2) of *GJB2*, and for the 342-kb deletion (*GJB6*-D3S1830) in *GJB6*. None of these 29 families exhibited the del(*GJB6*-D13S1830) mutation or any of the reported disease-causing mutations in *GJB2*. However, the *GJB2* variant of uncertain significance, c.499G>A (p.V167M), was present in one family in the heterozygous form [29].

In 2016, a panel of 116 HI genes (OtoSCOPE^®^ platform) was interrogated through targeted genomic enrichment and massively parallel sequencing. In 7 out of 10 families investigated (70%), 12 putatively pathogenic variants were identified in six NSHI genes (*CHD23, LOXHD1, MYO7A, SLC26A4, OTOF,* and *STRC*) [34]. Five of the 12 identified variants (41.6%) were novel, whereas the remaining seven variants have been shown to be involved in NSHI in populations outside of sub-Saharan Africa. All identified variants segregated with HI phenotype [34]. The prevalence of these newly identified mutations was assessed in a cohort of 57 non-syndromic hearing-impaired individuals from Cameroon. Only variants *OTOF* NM_194248.2:c.766-2A>G and *MYO7A* NM_000260.3:c.1996C>T (p.Arg666Stop) were found in 3 (5.3%) and 5 (8.8%) patients, respectively [28]. All these variants were in a heterozygous state in all cases and are thus unlikely to explain, alone, the cause of HI in these patients [28].

##### Keratitis–Ichtyosis–Deafness Syndrome

The coding exons (exon2) of *GJB2* of the two unrelated Cameroonian patients with KID syndrome were screened through direct sequencing. A pathogenic missense mutation, c.148G>A, resulting in a putative amino acid change from aspartic acid (GAC) to asparagine (AAC) in codon 50, p.Asp50Asn, was identified in the heterozygous form in both patients [30]. The “NM_004004.6:c.148G>A” mutation was not present in more than 180 unrelated individuals who were screened for recessive deafness mutations, nor in 60 healthy control persons of Cameroonian origin [30].

## 4. Discussion

To our knowledge, this systematic review is the first report summarizing data on the prevalence, etiologies and genetics of HI in Cameroon. Our study confirms the lack of nationwide studies evaluating the prevalence or the incidence of HI in Cameroon.

When using a threshold of 40 dB in adults and 35 dB in children, the population-based prevalence of HI in Cameroon is about 3.6% and increases with age. It is low at 1.1% in children, is up to 6.5% in adults and rises to a level of 14.8% in participants aged of 50 years or more. This is consistent with report from the WHO which estimated that the prevalence of disabling HI in sub-Saharan Africa is about 4.5% in the general population, 1.9% in children and 6.4% in adults [1]. HI prevalence varies greatly across studies in sub-Saharan Africa. Population-based prevalence of HI ranges from 5% in Mozambique to 9.1% in Sierra Leone and 18% in Uganda [45,46,47].

Several factors contribute to this prevalence variation across studies, including different study settings (community-, school- or hospital-based) and the use of different hearing test techniques [48]. When using PTA as the hearing test technique, different cut-offs used to diagnose HI would also contribute to prevalence variation across studies. Using a low cut-off such as 25 dB would identify milder HI and would produce a higher prevalence, whilst a pure tone cut-off at 40 dB would result in a lower prevalence as only moderate and severe HI would be identified [48]. The prevalence of HI in a population-based study from South Africa was estimated at 12.3% when using a cut off of 25 dB, while it was low at 4.6% when the cut off was 30 dB in children and 40 dB in adults [49]. Additionally, different mean ages of the study populations across studies will also lead to different prevalence across these studies, since HI prevalence was shown to increase with age [1]; therefore, the older the population, the higher the prevalence. The prevalence of HI in a tertiary health institution in Southwestern Nigeria was estimated at 7.5% in children (≤15 years old), whilst it was up to 34.5% in adults (>15 years old) [50].

Our study confirms that environmental and preventable factors play a major role in the etiology of HI in Cameroon, as is the case in other sub-Saharan African countries (Table 1). Environmental factors contribute to 65.5%, 45.7% and 37.5% of HI cases in Sierra Leone, Nigeria and Gambia, respectively [6,43,44]. Infectious diseases that are preventable by vaccination (meningitis, measles, rubella and mumps) were the leading environmental etiologies and represented 41.3% of cases, which is in line with findings from Sierra Leone and Gambia, where preventable infectious diseases were the leading environmental cause of HI and contributed to 44.7% and 35.1% of HI cases, respectively [6,43]. The substantial implication of preventable diseases in the etiology of HI in sub-Saharan Africa highlights the weakness of health care systems, and the need for sub-Saharan African countries to develop strategies to reduce the prevalence of HI that is due to preventable diseases. Strategies to fight HI in sub-Saharan Africa should include reinforcement of immunization programs, adequate equipment in health care facilities, and early diagnosis and proper management of HI-causing diseases.

In this review, impacted wax and age-related HI were identified as the second and the third most frequent causes of HI, respectively, after meningitis. They were also found to be important causes of HI in other sub-Saharan African populations. Impacted wax was highly prevalent in Nigerian children and was responsible for 53% of HI cases, whilst it was less prevalent but not negligible in Uganda and Zimbabwe, where it was implicated in 10% and 13% of HI cases, respectively [22]. This accentuates the important role of primary healthcare facilities in the prevention of HI, as impacted wax can be easily treated through primary healthcare services [32]. Presbycusis, also called age-related HI, can be defined as a progressive, bilateral and symmetrical sensorineural HI, due to age-related degeneration of inner ear structures [51]. It is a multifactorial disorder, with the implication of both environmental and genetic factors [51,52]. Presbycusis was implicated in 22.7% of HI cases in an elderly population from Nigeria [53].

As previously reported by other studies from sub-Saharan African countries, the etiology of HI remains unknown in approximately one-third of HI cases in the Cameroonian population (Table 1). This can be explained by the poor quality of medical records and limited clinical, biological, genetic and morphological (e.g., CT scan) investigational capacities in most settings [27]. The low contribution of genetic factors to HI was also reported in other sub-Saharan African countries (Table 1). This can be attributed to limited access to molecular screening in some sub-Saharan African countries, as many cases of HI classified as being of unknown origin might be of genetic origin [54]. There is therefore a need to intensify research on the genetics of HI in sub-Saharan Africa to establish a comprehensive list of the most frequent mutations associated with HI in these populations, as many cases of genetic HI cases presenting as sporadic cases will be classified as of unknown origin in the absence of molecular diagnosis [27].

Our study identified WS as the most prevalent syndromic HI in Cameroon, in line with another study performed on African subjects, in which 62% of black children with syndromic deafness from Southern Africa presented with WS [55]. WS is defined as an auditory–pigmentary disorder which affects the iris, hair and skin’s pigmentary deposits [7]. WS has four subgroups categorized according to the presence or absence of coexisting abnormalities [11]. In WS type 1, dystrophia canthorum is observed in every patient, while there is no dystrophia canthorum in WS type 2 [11]. In WS type 3, upper extremity abnormalities and the findings of type 1 are found [11]. In WS type 4, there are pigmentation abnormalities, Hirschsprung disease and the findings of type 2 [11]. All the cases reported in the Cameroonian population were type 2 WS. Type 1 WS was described in some African populations, including 18 cases from Kenya [56] and 31 cases from South Africa [57]. No genetic exploration was performed on cases of WS described in the Cameroonian population, however, the most common genes incriminated in the etiology of WS in previous studies include *PAX*, *EDRNB*, *SOX10*, *MITF* and *EDN3* [11,58,59,60,61].

This review confirms that *GJB2* and *GJB6* genes, the major contributors to NSHI in Caucasian and Asian populations [62,63,64,65,66,67], are not significant in NSHI in the Cameroonian population. Apart from the Ghanaian population, where the *GJB2* founder mutation p.R143W (c.427C˃T) was shown to be highly prevalent [17] and implicated in approximately 25% of familial cases and 8% of isolated cases of HI [18], our study is consistent with previous reports in other populations of African descent [68,69]. Mutations in *GJB6*, including the 342-kb deletion, *GJB6*-D13S1830, were not reported in cohorts of 182, 44 and 401 probands from South Africa, Nigeria and Ghana, respectively [18,69,70]. As in the Cameroonian population, disease-causing mutations in *GJB2* are also less prevalent or absent in populations from Kenya, Sudan, Uganda, Nigeria, Mauritania and South Africa [5,33,69,70,71,72].

Targeted gene sequencing identified putatively pathogenic variants in 7 out of 10 Cameroonian families (70%), and the prevalence of these variants was <10% in 82 probands from Cameroon and South Africa. This result highlights the low contribution of other known HI genes to NSHI in sub-Saharan African populations, and the promise of next generation sequencing (NGS) in finding variants associated with HI in these populations. Next generation sequencing techniques have demonstrated their efficacy in resolving HI cases across many populations. Targeted gene sequencing was able to identify pathogenic variants that co-segregate with HI in 46.7%, 52.9% and 60% of Japanese, Turkish and Chinese families, respectively, which had presented NSHI and which did not have pathogenic mutations in common HI genes [73,74,75]. Because of the high genetic heterogeneity of NSHI, NGS appears to be the most effective method to identify variants associated with non-syndromic deafness in African populations.

## 5. Strengths and Limitations

Of the 17 studies included in this review, only two prevalence studies were population-based, and they were both conducted in just one of the ten administrative regions of Cameroon. Therefore, we do not have a national estimate of the prevalence of HI in Cameroon. Additionally, in one of these two population-based studies, HI cases were diagnosed on a self-reported base, allowing a potential bias in the estimate of the prevalence of HI in that region.

However, the present review is, to the best of our knowledge, the first report assessing the burden of HI in Cameroon. This study has public health implications, since it will raise awareness among policy-makers and will help them develop strategies to reduce the burden of HI (including implementation of a newborn screening program, reinforcement of the immunization program and the improvement of equipment available at health facilities). Additionally, this review should encourage the funding of research projects that aim to identify gene variants associated with HI in the Cameroonian population, as our study showed that known HI genes do not have a significant role in HI in this population.

## 6. Conclusions

The prevalence of HI in Cameroon is similar to the WHO estimate of HI prevalence in sub-Saharan Africa. Environmental factors are the leading causes of HI in Cameroon, with meningitis, impacted wax and age-related disorder being the most common causes. Concerning HI of genetic origin, autosomal recessive inheritance is the most frequently observed inheritance pattern. NSHI is the most common clinical entity, and Waardenburg and Usher syndromes have been identified as the most frequent syndromic HI in the Cameroonian population. The high implication of diseases that are preventable by vaccination (meningitis, measles, rubella and mumps) in the etiology of HI highlights the need to reinforce the current national immunization program in Cameroon, and the very low implication of known HI genes in the Cameroonian population shown by our study highlights the need for next-generation sequencing techniques to identify novel variants that are associated with HI in populations of African descent.

## 7. Research Perspectives

There is an obvious need for population-based study at a national level to estimate the prevalence and incidence of HI in Cameroon. Additionally, whole genome sequencing data should be generated from non-syndromic hearing-impaired patients and analyzed in order to identify genes that contribute to NSHI in Cameroon. Functional studies should also be performed to understand the molecular mechanisms underlying the pathogenicity of HI in the Cameroonian population.

## Figures and Tables

**Figure 1 genes-11-00233-f001:**
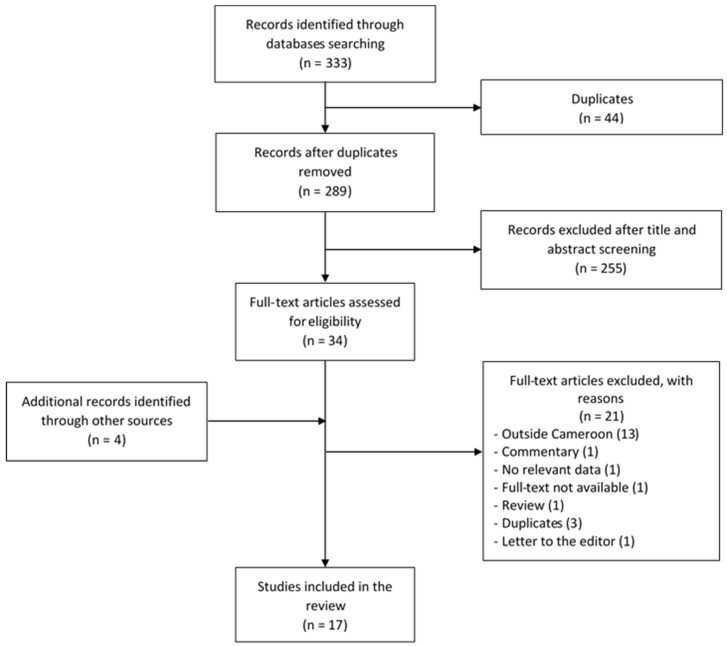
Flow chart of studies selection.

**Figure 2 genes-11-00233-f002:**
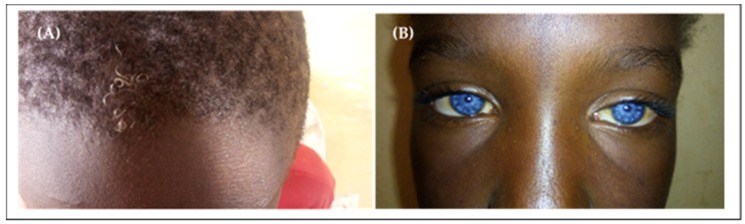
Illustration of some clinical signs found in Cameroonian patients with Waardenburg syndrome. (**A**) Premature white hair; (**B**) Sapphire-blue eyes (extracted from the study by Tingang Wonkam et al. [29]).

**Figure 3 genes-11-00233-f003:**
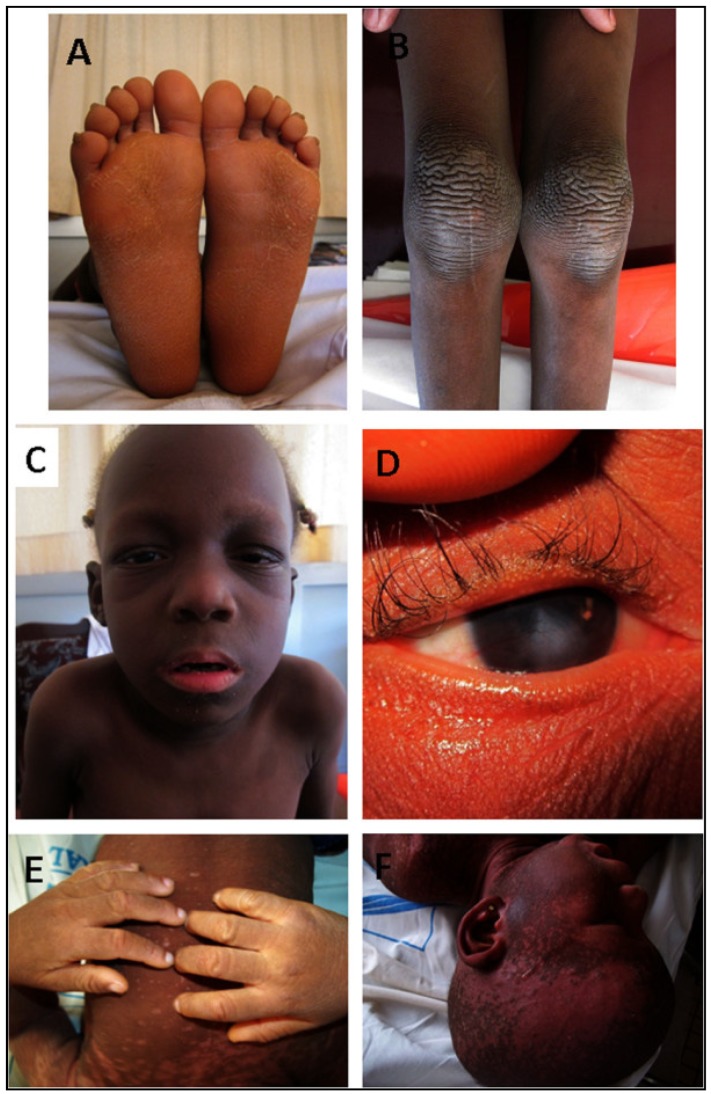
Illustrations of some clinical features of the two Cameroonian KID cases (Case 1; panels A–D; Case 2 panels E and F). (**A**) Keratoderma of the soles; (**B**) Rippled hyperkeratotic plaques on the knees; (**C**) Hypotrichosis of the eyelashes and eyebrows; (**D**) Mild vascularizing keratitis; (**E**) Hyperkeratosis of the hands; (**F**) Alopecia, hypotrichosis, ichthyosiform erythrokeratoderma (extracted from the paper by Wonkam et al. [30]).

**Table 1 genes-11-00233-t001:** General characteristics of the included studies.

First Author’s Name, Publication Year	Area	Regions	Study Setting	Study Design	Data Collection	Study Population	Male (%)	Mean Age (Years)	Age Range (Years)	Sample Size	Diagnosis Tool	Quality
Fokouo, 2015 [26]	urban	center region	hospital	case-control	prospective	patients followed up for HIV infection.	28.3	33.4 ± 7.7	15–49	180	PTA	high
Wonkam, 2013 [27]	urban and rural	7 regions	school and hospital	cross sectional	prospective	patients with childhood deafness	54.1	11 *	1–32	582	PTA and ABR	high
Lebeko, 2017 [28]	urban and rural	7 regions	schools and hospital	cross sectional	prospective	patients with non-syndromic hearing impairment of either putative genetic origin or unknown origin	NR	NR	NR	57	PTA	high
Djomou, 2016 [39]	urban	center region	hospital	cross sectional	retrospective	patients admitted at the ENT unit for sensorineural emergencies	NR	43.2 ± 17	16–66	22	PTA	low
Tingang Wonkam, 2019 [29]	urban and rural	8 regions	school and community	cross sectional	prospective	familial hearing impairment cases	45.2	18 ± 10.4	1–50	93	PTA and ABR	high
Trotta, 2011 [36]	rural	Far-North region	school	cross sectional	prospective	patients with prelingual hearing loss	74.4	NR	>5	70	PTA	moderate
Wonkam, 2013 [30]	NR	NR	school	Case report	prospective	patients suffering from KID syndrome	0	3.5 ± 2.12	2–5	2	PTA and ABR	high
Kuaban, 2015 [37]	urban	2 regions	hospital	longitudinal	prospective	MDR-TB (multidrug-resistant tuberculosis) patients, treated with a standardized 12-months regiment, including kanamycin.	51.3	33.7	17–68	150	PTA	moderate
Jivraj, 2014 [40]	rural	Northwest region	school	cross sectional	prospective	students at two schools, including a school for the deaf	58.2	11.8 ± 2.8	NR	320	NR	low
Bosch, 2014 [31]	urban and rural	7 regions	school and hospital	cross sectional	prospective	patients with deafness of either putative genetic origin or unknown origin and that were shown not to have mutation in *GJB2* gene	52	12.11	NR	75	PTA	high
Ferrite, 2017 [32]	rural	Northwest region	community	cross sectional	prospective	general population of the Fundong Health District, Northwest Cameroon	40.8	24.4	0–80+	3567	PTA and OAE	high
Bosch, 2014 [33]	urban and rural	7 regions	school and hospital	cross sectional	prospective	patients with deafness of either putative genetic origin or unknown origin	NR	NR	NR	180	PTA	high
Lebeko, 2016 [34]	urban and rural	NR	school and hospital	cross sectional	prospective	families with at least two individuals with ARNSHI who were negative for pathogenic variants in *GJB2* and *GJB6*	53.8	NR	NR	26	PTA	high
Chiabi, 2004 [41]	rural	East region	hospital	cross sectional	retrospective	patients admitted and treated for severe malaria	55.3	2.7	0–15	387	self-reported	low
Trébucq, 2018 [38]	urban	NR	hospital	longitudinal	prospective	MDR-TB (multidrug-resistant tuberculosis) patients, treated with a standardized 9-months regiment, including kanamycin	NR	NR	≥18	176	PTA	moderate
Cockburn, 2014 [42]	rural	Northwest region	community	cross sectional	prospective	people living in the Northwest region of Cameroon	43.3	NR	0–70+	18 878	self-reported	low
Noubiap, 2014 [35]	urban and rural	7 regions	schools and hospital	cross sectional	prospective	patients suffering from Waardenburg syndrome	50	12.2 ± 7	6–25	6	PTA	high

ABR, auditory brainstem response test; ARNSHI, autosomal recessive non-syndromic hearing impairment; ENT, ear nose and throat; KID, keratitis–ichthyosis–deafness; NR, not reported; OAE, otoacoustic emission test; PTA, pure tone audiometry; * The number shown here is the median.

**Table 2 genes-11-00233-t002:** Etiologies of hearing impairment in Cameroon and comparison to other African countries.

Country	Cameroon	Cameroon	Sierra Leone	Gambia	Ghana	Nigeria
Year of publication	2013	2017	1991	1985	2019	1982
Reference	[27]	[32]	[6]	[43]	[18]	[44]
Number of patients	582	127	354	259	1104	298
Hereditary	14.8%	0.8%	–	8.1%	21.3%	13.1%
Meningitis	34.4%	–	23.9%	31.7%	3.9%	11%
Impacted wax	–	31.5%	–	–	–	–
Age-related HI	–	22.8%	–	–	–	–
Noise-induced HI	–	1.5%	–	–	–	–
Measles	4.3%	–	4.1%	1.9%	0.9%	13%
Rubella	0.5%	–	–	1.5%	0.2%	2%
Mumps	2.1%	–	16.7%	–	0.5%	3%
Ototoxicity	6%	–	20.8%	–	–	9%
Other	5.3%	6.4%	–	2.3	13.1%	7.7%
Unknown	32.6%	37%	34.8%	54.4%	60.1%	41.2%

HI, hearing impairment.

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
