# Peer review of "Hearing Impairment Overview in Africa: the Case of Cameroon"

_genes, 2020, doi:10.3390/genes11020233_

Round 1

Reviewer 1 Report

the paper is a good systematic review on hearing loss in Cameroon, only limited by the fact that only 17 papers were included. 

Author Response

We are grateful for the positive comments. We trust to have included all the available articles on hearing impairment in Cameroon.

Reviewer 2 Report

It is very encouraging to see that the study of hearing loss in sub-Saharan Africa is rapidly advancing. We are now at the stage where considerable knowledge has been gained regarding prevalence, aetiology and genetic basis of non-syndromic disorders. The authors have written a high quality review that summarises the research carried out with children and adults in Cameroon on hearing loss prevalence and aetiology. A strength of the review is the detailed consideration of the genetic causes of hearing loss so far noted. The authors highlight the differences in known genetic aetiologies of hearing impairment in African versus Western populations. Overall I consider this manuscript a very useful addition to the literature on hearing loss in sub-Saharan Afric and commend the authors for their thoughtful work. My minor comments for possible correction etc. are listed below.

Author Affiliations: Please correct typo -- should be "Centre for Heart Rhythm Disorders" not Hearth

Abstract: Should be "Reference lists of included studies ..."; "age-related process" is a rather clumsy description. Why not just "presbycusis" or "age-related disorder"?

Introduction: Should be "of whom 34 million" not "of which".

Materials and Methods: Table 1 The order of the first column title is incorrect, it should be "First author's name, publication year"; No where in the review do the authors define what they mean by "mild", "moderate" etc. hearing loss. Can the authors define these terms in relation to dB HL levels and give a reference to the classification system they have chosen to use?; "In addition, all the 28 cases ... presented with HI [40]." This sentence is rather redundant. All the children were attending a school for the deaf, so obviously they all would present with hearing loss. This sentence needs alteration of just delete it; introduce the full label for KID syndrome the first time this is mentioned [keratitis-ichthyosis-deafness syndrome]; use full label for this syndrome for 3.6.3.2 title;.

Discussion: "... when the cut off was 30dB in children ..." should be 30 dB [add the space].

References: [12] has no authors listed - please add; [19] no journal volume number or page numbers are shown; [40] Mbingo is a place and should be capitalised = Mbingo not "mbingo"; [41] Why is the full journal title capitalised? please correct; [43] is a duplicate. This reference is already listed as reference [33]; [44] is a duplicate. This reference is already listed as reference [35]; [62] Why is the full article title capitalised? Please correct. Please check all references are correctly linked to citations in your final revision. For reviews it is even more important than usual that references are accurate.

Author Response

It is very encouraging to see that the study of hearing loss in sub-Saharan Africa is rapidly advancing. We are now at the stage where considerable knowledge has been gained regarding prevalence, aetiology and genetic basis of non-syndromic disorders. The authors have written a high quality review that summarises the research carried out with children and adults in Cameroon on hearing loss prevalence and aetiology. A strength of the review is the detailed consideration of the genetic causes of hearing loss so far noted. The authors highlight the differences in known genetic aetiologies of hearing impairment in African versus Western populations. Overall I consider this manuscript a very useful addition to the literature on hearing loss in sub-Saharan Afric and commend the authors for their thoughtful work. My minor comments for possible correction etc. are listed below.

We are grateful for the positive comments.

Author affiliations

Reviewer’s comment 1: Please correct typo -- should be "Centre for Heart Rhythm Disorders" not Hearth.

Authors’ response 1: Thank you dear reviewer for your comment. This has been corrected, and the section Author Affiliations/L9 now reads: “Centre for Heart Rhythm Disorders, South Australian Health and Medical Research Institute (SAHMRI)”.

Abstract

Reviewer’s comment 2: Should be "Reference lists of included studies ..."

Authors’ response 2: Thank you for your comment. Revision has been made, and the section Abstact/L24-25 now reads: “…Reference lists of included studies were also scrutinized, and data were summarised narratively…”

Reviewer’s comment 3: "age-related process" is a rather clumsy description. Why not just "presbycusis" or "age-related disorder"?

Authors’ response 3: Thank you dear reviewer for your comment, the authors agree with the reviewer. The word “process” has been replaced by “disorder”, and the section Abstract/L28-29 now reads: “…Environmental factors contribute to 52.6 to 62.2% of HI cases; meningitis, impacted wax and age-related disorder being the most common ones…”. In addition, the section Conclusions/L418-419 now reads: “…Environmental factors are the leading causes of HI in Cameroon, with meningitis, impacted wax and age-related disorder being the most common causes…”.

Introduction

Reviewer’s comment 4: Should be "of whom 34 million" not "of which".

Authors’ response 4: Thank you for your comment. This has been corrected. The section Introduction/L46-48 no reads: “…The World Health Organisation (WHO) estimated in 2018 that… of whom 34 million are children…”.

Materials and Methods

Reviewer’s comment 5: Table 1 the order of the first column title is incorrect, it should be "First author's name, publication year".

Authors’ response 5: Thank you for your comment dear reviewer. The authors agree with the reviewer. The first column title of Table 1 has been changed as requested to “First author’s name, publication year”.

Reviewer’s comment 6: No where in the review do the authors define what they mean by "mild", "moderate" etc. hearing loss. Can the authors define these terms in relation to dB HL levels and give a reference to the classification system they have chosen to use?

Authors’ response 6: Thank you dear reviewer for your comment. We have now added the classification of Hearing impairment based on the dB levels. The section Introduction/L51-52 now reads: “According to WHO, HI can be classified as mild, moderate, severe or profound, when the pure tone average ranges from 26 to 40 dB, 41 to 60 dB, 61 to 80 dB, or is over 81 dB, respectively [1]…”.

Reviewer’s comment 7: In addition, all the 28 cases ... presented with HI [40]." This sentence is rather redundant. All the children were attending a school for the deaf, so obviously they all would present with hearing loss. This sentence needs alteration of just delete it.

Authors’ response 7: Thank you for your comment. The authors agree with the reviewer. The sentence in section Results/Aetiologies of Hearing Impairment in Cameroon/L185-186: “…In addition, all the 28 cases of probable congenital rubella syndrome (CRS) presented with HI [40].” has been removed.

Reviewer’s comment 8: introduce the full label for KID syndrome the first time this is mentioned [keratitis-ichthyosis-deafness syndrome]; use full label for this syndrome for 3.6.3.2 title.

Authors’ response 8: Thank you for your comment. The full label for KID syndrome has been introduced the first time this is mentioned. The section Results/Hearing Impairment of Genetic Origin/Clinical Patterns/L241 now reads: “Two cases of keratitis-ichthyosis-deafness (KID) syndrome were reported…”. In addition, the full label for this syndrome has been used for 3.6.3.2 title, in the section Results/Hearing Impairment of Genetic Origin/Genes Variants and Hearing Impairment/Keratitis-Ichtyosis-Deafness Syndrome/L302.

Discussion

Reviewer’s comment 9: "... when the cut off was 30dB in children ..." should be 30 dB [add the space].

Authors’ response 9: Thank you for your comment. The space has been added, and the section Discussion/L328 now reads: “…when the cut off was 30 dB in children…”.

References

Reviewer’s comment 10: [12] has no authors listed - please add.

Authors’ response 10: Thank you dear reviewer for your comment. The authors’ name for reference [12] have been added. The section Reference/L475 now reads: “Hilgert, N.; Smith, R.J.H.; Van Camp, G. Forty-six genes causing nonsyndromic hearing impairment…”

Reviewer’s comment 11: [19] no journal volume number or page numbers are shown.

Authors’ response 11: Thank you for comment dear reviewer. The reference [19] has been retrieve from the website GeneReviews®. The URL of the website has been added, and the section Reference/L495-498 now reads: “Shearer, A.E.; Hildebrand, M.S.; Smith, R.J. Hereditary Hearing Loss and Deafness Overview. In: Adam, M.P., Ardinger, H.H., Pagon R.A., et al., editors. GeneReviews® [internet]. University of Washington, Seattle, WA, USA, 1993-2020. 1999 Feb 14 [Updated 2017 Jul 27]. Available from: https://www.ncbi.nlm.nih.gov/books/NBK1434/ (accessed on Feb 14, 2020)”

Reviewer’s comment 12: [40] Mbingo is a place and should be capitalised = Mbingo not "mbingo".

Authors’ response 12: Thank you for your comment. This has been corrected. The reference [40] in the section Reference/L556 now reads: “manifestations of congenital rubella syndrome in Mbingo…”

Reviewer’s comment 13: [41] Why is the full journal title capitalised? please correct.

Authors’ response 13: Thank you dear reviewer for your comment. The capitalization has been removed. The reference [41] which is now reference [42] in the section Reference/L560 now reads: “…Region, Cameroon. Health Sciences and Diseases …”

Reviewer’s comment 14: [43] is a duplicate. This reference is already listed as reference [33].

Authors’ response 14:  Thank you for your comment dear reviewer. The authors agree with the reviewer. The reference [43] which was a duplicate has been replace by reference [33].

Reviewer’s comment 15: [44] is a duplicate. This reference is already listed as reference [35].

Authors’ response 15: Thank you dear reviewer for your comment. The reference [44] which was a duplicate has been replaced by reference [35].

Reviewer’s comment 16: [62] Why is the full article title capitalised? Please correct.

Authors’ response 16: Thank you for your comment. The full capitalization of the article tile has been removed. The reference [62] which is now reference [54] in the section Reference/L582 now reads: “Fraser, G.R. Profound Childhood Deafness. J. Med. Genet. 1964, 1, 118–151.”

Reviewer’s comment 17: Please check all references are correctly linked to citations in your final revision. For reviews it is even more important than usual that references are accurate

Authors’ response 16: Thank you dear reviewer for your suggestion. We have cross checked all references and citations.

Other revisions:

Section Authors’ name/L4: Author’s name “Jean Jacques N. Noubiap” has been changed to “Jean Jacques Noubiap”. The “N.” has been removed for citation purpose, at the request of the author.

Section Authors’ affiliations/L7: The abbreviated author’s name “E.C.” has been changed to “E.R.C.”

Section Materials and Methods/Assessment of Methodological Quality/L122: “…Two investigators (EWT and EC)…” has been changed to “…Two investigators (EWT and ERC)…”.

Section Authors’ contribution/L445-446: The abbreviated author’s name “EC” has been changed to “ERC”.

The section Introduction/L44 now reads: “Hearing impairment (HI) is considered disabling when the loss of hearing is greater than 40_dB”. A space has been added between “40” and “dB”.

The section Results/ Audiometric Characteristics of Hearing Impairment in Cameroon/L166-167: “…profound HI (≥80 dB)…” has been changed to “…profound HI (≥81 dB)…”

Supplementary materials

The title has been changed from “Hearing Impairment Overview in sub-Saharan Africa, the Case of Cameroon” to “Hearing Impairment Overview in Africa: the Case of Cameroon”.

The author’s name “Emile Chimusa” has been chanted to “Emile R. Chimusa”.
